# Management of Vascular Access in the Setting of Percutaneous Mechanical Circulatory Support (pMCS): Sheaths, Vascular Access and Closure Systems

**DOI:** 10.3390/jpm13020293

**Published:** 2023-02-06

**Authors:** Andrea Sardone, Luca Franchin, Diego Moniaci, Salvatore Colangelo, Francesco Colombo, Giacomo Boccuzzi, Mario Iannaccone

**Affiliations:** 1Division of Cardiology, San Giovanni Bosco Hospital, ASL Città di Torino, 10100 Turin, Italy; 2Division of Vascular Surgery, San Giovanni Bosco Hospital, ASL Città di Torino, 10100 Turin, Italy

**Keywords:** vascular management, mechanical circulatory support, protect-PCI, cardiogenic shock

## Abstract

The use of percutaneous mechanical circulatory support (pMCS), such as intra-aortic balloon pump, Impella, TandemHeart and VA-ECMO, in the setting of cardiogenic shock or in protect percutaneous coronary intervention (protect-PCI) is rapidly increasing in clinical practice. The major problem related to the use of pMCS is the management of all the device-related complications and of any vascular injury. MCS often requires large-bore access, if compared with common PCI, and for this reason the correct management of vascular access is a crucial point. The correct use of these devices in catheterization laboratories requires specific knowledge such as the correct evaluation of the vascular access performed, when possible, with advance imaging techniques in order to choose a percutaneous or a surgical approach. In addition to conventional transfemoral access, other types of access, such as transaxillary/subclavial access and the transcaval approach, have emerged over the years. These other approaches require advanced skills of the operators and a multidisciplinary team with dedicated physicians. Another important part of the management of vascular access is the closure systems used for hemostasis. Currently, two types of devices are typically used in the lab: suture-based or plug-based ones. In this review we want to describe all these aspects related to the management of vascular access in pMCS and describe, finally, a case report from our center’s experience.

## 1. Introduction

Over the years, percutaneous mechanical circulatory support (pMCS) has been increasingly used in the setting of cardiogenic shock (CS) or in high-risk percutaneous coronary intervention (PCI) [1,2,3]. The main goal of pMCS is to unload the left ventricle in order to reduce myocardial oxygen demand, maintaining systemic and coronary perfusion in the setting of both cardiogenic shock (CS) and high-risk PCI (HR-PCI) [4,5]. Despite the progress made throughout the years, the management of vascular access and complications is still of paramount interest to improve clinical outcomes. Noteworthy, the rates of life-threatening or severe bleeding complications are more common in the setting of cardiogenic shock compared with HR-PCI [6]. Vascular management with pMCS is different from other procedures involving the use of large-bore sheaths (transcatheter aortic valve replacement or endovascular aortic endoprosthesis replacement), as an accurate preliminary assessment with computed tomography scan (CT scan) is not always possible in the setting of CS. The aim of this review is to provide a thorough evaluation of sheaths, vascular access management and closure systems in different clinical scenarios.

## 2. Mechanical Circulatory Support and Large-Bore Sheaths

The sheath’s main purpose is to get access to the vessel lumen while avoiding loss of blood by using a hemostatic valve. Despite technology advancement in reducing sheath dimensions, the main issue related to the use of pMCS is still the need of large-bore dimensions.

### 2.1. Intra-Aortic Balloon Pump

The intra-aortic balloon pump (Arrow IABP, Winston-Salem, NC, USA) is one of the oldest and most often used pMCS devices. It is a counterpulsation device composed of an inflatable balloon attached to a double-lumen catheter and a pump console. The introducer sheath is used to place the balloon into the descending aorta. Its dimensions range from 7 Fr to 9 Fr with a length of 15 cm. It is mainly used for CS due to acute myocardial infarction mechanical complications, because its hemodynamic support is of limited with a flow of 0.5 L/min.

### 2.2. Impella

The Impella device (Abiomed, Danvers, MA, USA) is a microaxial-flow pump that produces nonpulsatile flow from the left ventricle into the ascending aorta. Three different devices are currently available: *Impella CP* (14 Fr pump motor, maximum flow rate 4.3 L/min), *Impella 5.0* (21 Fr pump motor, maximum flow rate 5.0 L/min), and *Impella 5.5* (19 Fr pump motor, maximum flow rate 6.2 L/min). Devices are inserted either percutaneously (Impella CP) into the femoral or axillary artery or by surgical cutdown (Impella 5.0 and Impella 5.5) into the femoral artery or other alternative arteries.

### 2.3. TandemHeart

The TandemHeart (CardiacAssist Inc, Pittsburgh, PA, USA) is a continuous-flow centrifugal extracorporeal assist device, withdrawing oxygenated blood from the left atrium and returning it to the femoral artery. The inflow cannula is inserted percutaneously through the femoral vein and advanced into the left atrium. The femoral vein sheath is 21 Fr, and the femoral artery outflow sheath ranges from 15 Fr to 19 Fr.

### 2.4. VA-ECMO

Extracorporeal membrane oxygenation (ECMO) has been increasingly used in the setting of CS or in protect percutaneous coronary intervention (protect-PCI). The veno-arterial ECMO (VA-ECMO) consists of a centrifugal flow pump, a controller, a heat exchanger, a membrane oxygenator, and venous inflow/arterial outflow cannulas. Patients can be cannulated through the femoral artery and femoral vein, or through surgical isolation using central cannulation. The femoral vein sheath is 18 Fr while the femoral artery sheath ranges from 15 Fr to 23 Fr.

## 3. Vascular Access

Various access sites can be obtained to secure a safe pMCS. The most used is the common femoral artery (CFA). Other options are the transaxillary/subclavian artery and the transcaval approach.

### 3.1. Transfemoral Access

This is considered the preferred route in the majority of the procedures because most interventional cardiologists are very familiar with the transfemoral technique due to their experience with percutaneous coronary intervention. The CFA bifurcates into the superficial and the deep femoral artery; the correct puncture site should be at the level of the femoral bone head Figure 1 (ideally in the middle part of it) to have a compressible spot and to decrease the risk of retroperitoneal hemorrhage.

Puncture sites not located in the CFA were associated with a higher rate of postcatheterization local vascular complications [7,8].

The most common complications that occurred are highlighted below:Pseudoaneurysm formation (1–6%)Arteriovenous fistula formation (<1%)Hematoma formation (6–10%)Venous thrombosisPericatheter clotVessel laceration (<1%)Acute vessel closure (<1%)

A puncture below the femoral bifurcation increases the risk of puncture in small vessels and is relatively noncompressible due to the lack of sufficient support. 

It is important to acknowledge two different settings: high-risk PCI (or protect-PCI) and cardiogenic shock. The former is often a stable situation in which preoperative planning with access evaluation through angio computed tomography scan (angio CT scan) of inferior vessels is mandatory. The latter is an emergency situation in which adequate preliminary planning is not always possible. In this situation the puncture should be performed as follow:*Fluoroscopy:* CFA position can be determined by using anatomic landmarks with pulsation and fluoroscopic guidance. The puncture should be performed with consideration of a “safe zone” in the segment of the CFA extending between the inferior epigastric artery and the distal portion of the CFA, ideally 1 cm above the femoral bifurcation, corresponding to the space between the inferior edge and the middle part of the femoral head. This approach might avoid the puncture below the bifurcation (occurring below the middle third of the femoral head in 95% of patients [9]) in order to avoid pseudoaneurysm and it might avoid even puncture below the emergency of inferior epigastric (occurring usually above the middle third of femoral head).In order to minimize and avoid complications occurring after the puncture, it is mandatory to identify landmarks for the dermotomy and arteriotomy site.The inguinal ligament is the anatomical landmark that separates the external iliac artery from the CFA.

A landmark to identify the inguinal ligament on the skin is the line drawn between the anterior superior iliac spine and the pubic symphysis; the transfemoral puncture should be performed always below this line.

2.*Ultrasound:* Despite the fact that current guidelines recommend [10] the use of ultrasound in the cardiac catheterization laboratory, its use remains infrequent. The real advantage of ultrasound is the high probability of correct puncture in the CFA and the possibility of avoiding puncture in segments of the vessel with atherosclerotic or calcified plaques of the anterior wall. However, to enhance the efficacy of ultrasound-guided puncture, a pre-evaluation of angiographic landmarks is required. To perform an ultrasound-guided puncture correctly and safely, it is necessary to follow the following steps:
Initial fluoroscopy identification of the lower edge of the femoral head with a hemostat or with a scissor Figure 2A and marking of this position as a landmark with a sterile marker Figure 2B.Once a safety zone with fluoroscopy has been obtained, the ultrasound probe should be positioned above the line marked on top of the femoral head to visualize and identify the bifurcation of the CFA Figure 2C.Under ultrasound guidance, local anesthetic should be injected.The next step is to insert the needle in the middle part of the probe with an angulation of 45°.The last step is the most challenging for beginners, with potential mistakes, such as changing the ultrasound position on the skin or changing the beam angle, potentially leading to high femoral puncture. The closer the needle entry point is to the ultrasound probe, the steeper is the angle required to triangulate the position.Angiography assessment should be performed immediately after sheath insertion.

3.*Micropuncture assess:* The rationale is to perform a small needle puncture with a smaller sheath in order to evaluate the accuracy of the site in CFA and the successful insertion of bigger sheaths. There are dedicated kits on the market to perform this kind of puncture; however, there are always several steps to follow:
The micropuncture needle (21 gauge) is used to cannulate the vessel; puncture could be fluoroscopic or ultrasound-guided.The Seldinger technique is used. A dedicated microwire (0.018″) is inserted inside the needle to get inside the vessel; then the needle can be removed.The dilator/introducer sheath (3 or 4 Fr) is railroaded over the micropuncture wire.The central dilator and the wire are removed, leaving the sheath in the vessel.Angiography evaluation might be performed to evaluate the site of the puncture.If the micropuncture site is correct, a standard wire (0.035″) can be inserted inside the sheath.Leaving the wire inside the vessel, it is possible to remove the micropuncture sheath and railroad a standard sheath over the wire.

4.*Angiography-guided femoral puncture:* A secondary access is necessary (radial or femoral) for the possibility of reaching the proximal part of the CFA to perform digital subtraction angiography (DSA) and road mapping Figure 3 to guide the puncture.

5.*Surgical:* This approach is usually preferred in select cases, such as extremely obese patients. The surgical approach could be performed to expose the artery and the subsequent “de visu” puncture Figure 4. Moreover, a vascular graft can be anastomosed to the CFA with the insertion of a short sheath in the conduit.This approach includes six different steps:-Skin incision on the groin Figure 5A-Exposure of CFA and the femoral bifurcation Figure 5B-Arteriotomy of CFA-Anastomosis between CFA and a vascular graft Figure 6A-Tunnellization of vascular graft under the groin skin Figure 6B-Insertion of the sheath inside the vascular graft Figure 6C

### 3.2. Transaxillary/Subclavian Access

The sheath is introduced through the axillary and subclavian arteries into the aorta. The normal caliber of the axillary artery ranges from 6 to 7 mm, allowing the insertion of sheaths and catheters until 18 Fr. The axillary artery has been shown to be an acceptable alternative access site for pMCS if conventional access cannot be used [11,12].

Anatomically, the axillary artery is divided into three segments:The first segment is between the lateral margin of the first rib and the medial border of the pectoralis minor muscle.The second segment is above the pectoralis minor muscle.The third segment is between the lateral border of the pectoralis minor muscle and the inferior border of the teres major muscle.

The access could be percutaneous Figure 6 or with surgical cutdown. Percutaneous access should be performed at the distal edge of the first segment or the proximal end of the second segment with the needle crossing the pectoralis minor muscle in order to obtain a compressible site corresponding with the second rib. Surgical cutdown is usually performed with a vascular graft anastomosed to the vessel with the insertion of a short sheath in the conduit. Alternatively, it could be performed with surgical exposure of the vessel.

### 3.3. Transcaval Access

The first transcaval access was performed in 2013 by Greenbaum et al [13]. during TAVR. It might be considered a suitable approach especially for pMCS in protect-PCI without a feasible femoral or axillary approach. This innovative solution bypasses the iliofemoral arteries through the femoral and iliac veins to create a sheath-mediated channel between the inferior vena cava (IVC) and the abdominal aorta. Transcaval access candidacy should involve a multidisciplinary heart team and pre-procedural CT scan with thin slice reconstruction to evaluate the anatomy of each patient. The evaluation is important for identifying the most suitable segment of the abdominal aorta (free of calcium) near the vena cava and without the interposition of other structures such as the small-bowel intestine. If possible, the puncture should be performed away from the renal artery and the vein and away from the aortoiliac bifurcation. After the index procedure, transcaval closure is performed with a nitinol occluder device, usually an Amplatzer ductal occluder 1 (Abbott Vascular, Santa Clara, CA) across the aortotomy.

## 4. Vascular Closure Systems

The expansion of structural and complex endovascular procedures has made large-bore access management of paramount importance to enhance the safety and efficacy of these procedures. With this rise, closure device utilization has substantially increased. Currently, two types of devices are mainly used in the lab: suture-based and plug-based ones. 

### 4.1. Suture-Based Devices

Suture-based devices have been reported to reduce mortality, the need for blood transfusions, and infections, finally translating into shorter admissions after interventions compared with surgical approaches [14]. The ProGlide (Abbott Vascular, Santa Clara, CA) device is the most widely used suture-based device. It utilizes a polypropylene monofilament loaded on suture needles for arteriotomy closure and is indicated for femoral arterial closure ranging from 5 Fr to 21 Fr and venous access sizes ranging from 5 Fr to 24 Fr. For arteriotomy greater than 8 Fr, two ProGlide systems should be used before sheath insertion, mostly by creating a figure-of-eight stitch [15].

Whereas the common femoral artery should be suitable for percutaneous closure, there are a few contraindications to its use: anterior wall or circumferential calcification, aneurysm, access diameters < 5 mm and relatively important obesity. The most common complication is bleeding. While oozing at the site is acceptable and may be controlled by manual compression, significant bleeding prior to the guidewire removal could be tackled by a second device deployment. Other complications may also occur, such as vascular closure failure, vascular occlusion, thrombosis and lymphoceles [16], but they are not very frequent.

Several studies analyzed in a systematic review of the Cochrane database on vascular closure devices for hemostasis of the femoral arterial puncture site have already shown a low incidence of major complications and have high success rates with these devices [17].

### 4.2. Plug- and Patch-Based Devices

The most recent devices are the collagen-based Manta closure system (Essential Medical, Exton, PA, USA) and the nitinol patch-based InClosure VCD system (InSeal Medical Ltd., Caesarea, Israel). These two options could be used as a single solution, while other plug-based closure devices such as Angio-Seal (Terumo Corporation, Tokyo, Japan) could be used in combination with a single ProGlide in a hybrid approach. The Manta device is specifically designed to close arteriotomy sites up to 25 Fr. Currently, Manta is available in two sizes: a 14 Fr designed for 10–14 Fr sheaths with a maximum outer diameter of 18 Fr and an 18 Fr device designed for 15–18 Fr arteriotomies with a maximum outer diameter of 25 Fr. The plug is slowly resorbed over 6 months, allowing for re-access with the advantage of having a radiographically visible marker to guide future access. A large pivotal study demonstrated rapid hemostasis with few complications using the Manta device [18]. In this analysis, hemostasis was reached in 94% of the patients with only one major vascular complication and no minor bleeding according to the VARC-2. The novelty of the device is that it does not require anticipated closure and can maintain the access throughout the entire deployment. The InClosure (InSeal Medical Ltd., Caesarea, Israel) is a patch-based closure device designed to reach hemostasis for 14 to 21 Fr. It consists of a biodegradable membrane on a nitinol frame that adapts to the vessel after deployment. Early data showed good efficacy and safety of the device [19]. The advantages of the closure system are the wide range of compatibility of sheath and vessel sizes, no need for pre-closure and the possibility of re-access for future interventions.

## 5. Decisional Algorithm: Where, When and Which Device

In this paper we would to propose our decisional algorithm Figure 7 in three different settings.

Cardiogenic shock and HR-PCI are different scenarios, and the same approach cannot be used in both. In the first case we are dealing with an emergency setting with hemodynamic instability, so the timing of intervention is crucial. In the second case it is possible to achieve an optimal evaluation of the patient.

We describe three different scenarios:
*Cardiogenic shock pre-PPCI:* Time is muscle. After basal coronary angiography, in order to place a pMCS the use of DSA would probably be the faster option Figure 8.Digital subtraction angiography could be obtained using the index access (radial access or CFA contralateral). From this angiography it is possible to obtain information related to the bifurcation site, such as stenosis of the femoral axis. With the use of road mapping it is possible to perform a safe and correct puncture.In case of severe disease of the femoral route, the only option is IABP (with a minimum diameter of 3 mm) or a primary PCI without support (in case of obstructive disease of the femoral axis).If the femoral route is feasible in patients older then 70 y, IABP is the best option, with subsequent hemostasis with manual compression (after ACT evaluation) sometime after the index procedure.If the patient is younger than 70 y, Impella CP seems to be the best option to manage the instability situation.The puncture should be performed with DSA road mapping, ultrasound or micropuncture.Management of vascular access is crucial in this setting, and is it possible to achieve the best hemostasis using 2 Proglide, 1 Proglide and 1 AngioSeal or Manta.*HR-PCI:* This is often a stable situation, so a full evaluation of the access with angio CT is recommended. If the femoral route is available in patients older than 75 y, the use of IABP is often the best option; the puncture should be performed with DSA road mapping or ultrasound, and hemostasis can be obtained with a single Proglide or with manual compression. If the patient is younger than 75 y, Impella CP or VA-ECMO could be the best option. In the first case, puncture should be performed (using angio CT) with ultrasound, and the management of vascular access is easily obtained with 2 Proglide, 1 Proglide and 1 AngioSeal or Manta.In the second case the surgical approach is often necessary. If the transfemoral route is not available, PCI without support could be an option for patients older than 75 y. Otherwise, the surgical approach (surgical cutdown of femoral access, TA or TC) is a feasible option with the support of a cardiac or vascular surgeon.*Cardiogenic shock post-PPCI:* In this situation we are often dealing with rapid hemodynamic deterioration a few hours or days after primary PCI, so it is important to offer the best option to support it.In patients older than 75 y, IABP is probably the best option. Otherwise, in patients younger than 75 y Impella 5.5 or VA-ECMO with surgical management is recommended Figure 9.


## 6. Conclusions

The use of percutaneous mechanical circulatory support (pMCS) is rapidly increasing in clinical practice for both cardiogenic shock and HR-PCI, and vascular management is still an important problem for interventional cardiologists.

In this review we describe our center’s experience and decisional algorithms in all possible settings.

## Figures and Tables

**Figure 1 jpm-13-00293-f001:**
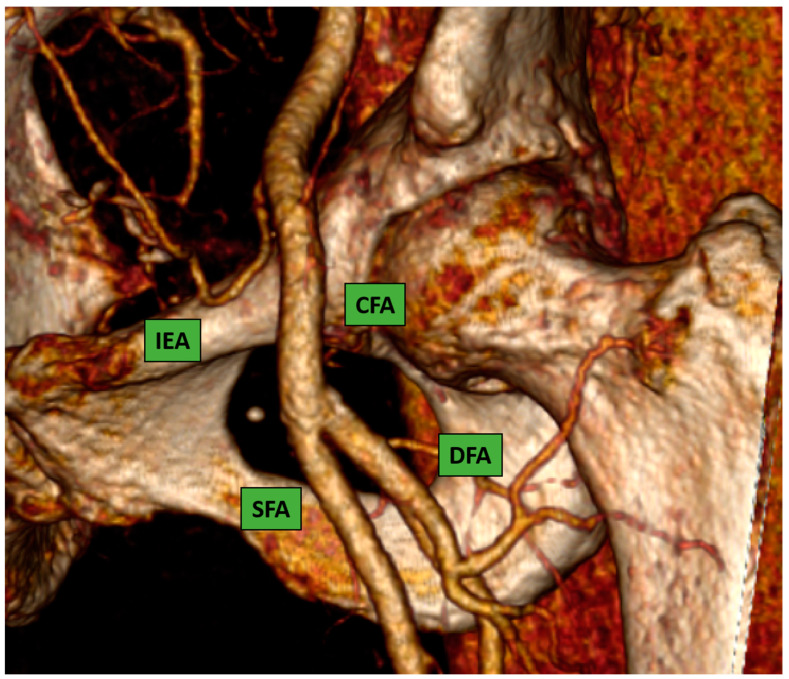
Common femoral artery and bifurcation into superficial and deep femoral artery. 3D reconstruction image from computed-tomography scan (CT-scan).

**Figure 2 jpm-13-00293-f002:**
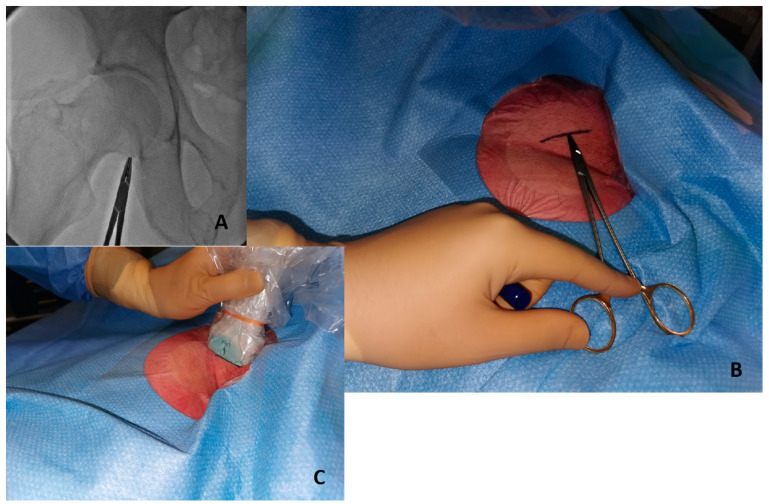
Initial step to perform ultrasound-guided puncture. (**A**) Identify lower edge of femoral edge with fluoroscopy and a metal hemostat. (**B**) Write a sign in the skin with a marker in the correspondent point. (**C**) The probe positioned above the marked line.

**Figure 3 jpm-13-00293-f003:**
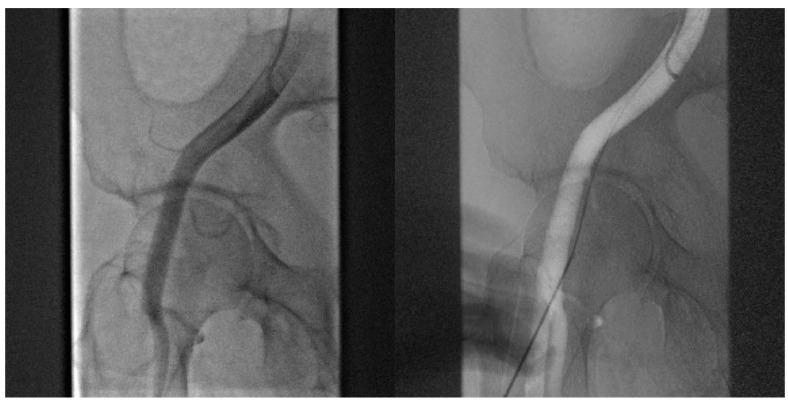
DSA allow to puncture in the correct site of CFA avoiding bifurcation and other small vessels.

**Figure 4 jpm-13-00293-f004:**
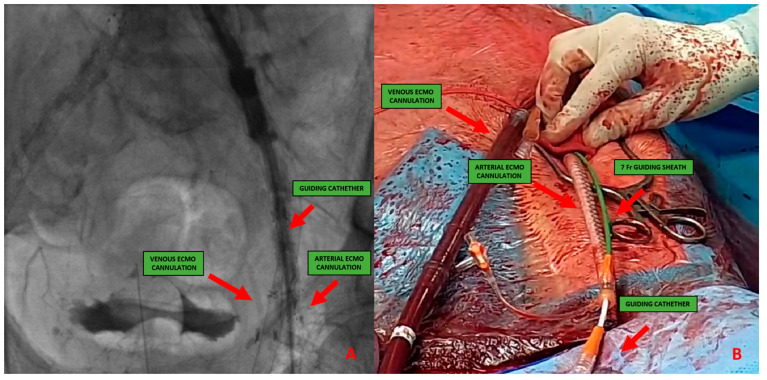
(**A**) 51-year-old man with history of: diabetes, chronic obstructive pulmonary desease (COPD), mild chronic kidney disease (GFR 56 mL/min) and peripheral artery disease (PAD) with previous percutaneous transluminal angioplasty (PTA) with stent implantation in the right external iliac artery, left iliac artery, left common femoral artery and subsequently aorto-bifemoral bypass surgery, referred to our center for HR-PCI A. Angiography evaluation after sheaths insertion. (**B**) Surgical cutdown with VA-ECMO cannulation and subsequently insertion of 7 Fr guiding sheats.

**Figure 5 jpm-13-00293-f005:**
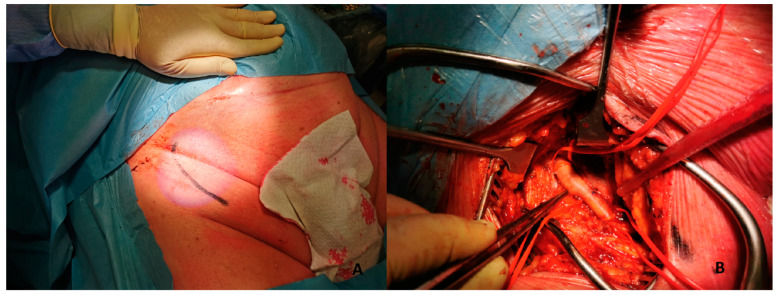
(**A**). Skin incision on the groin. (**B**). Exposure of common femoral artery.

**Figure 6 jpm-13-00293-f006:**
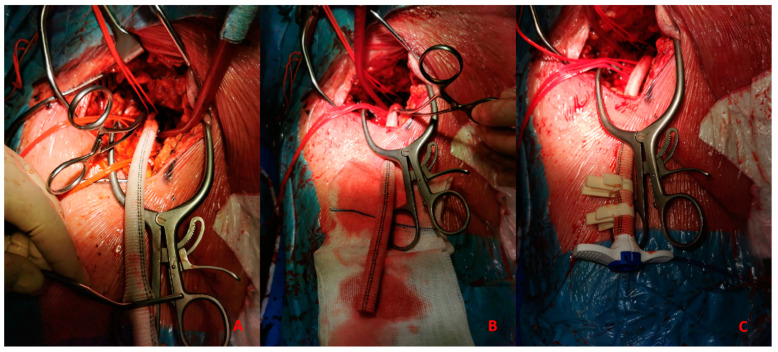
(**A**). Vascular graft and common femoral artery anastomosis. (**B**). Tunnellization of vascular graft under the skin. (**C**). Insertion of Impella 5.0 sheath in the vascular graft.

**Figure 7 jpm-13-00293-f007:**
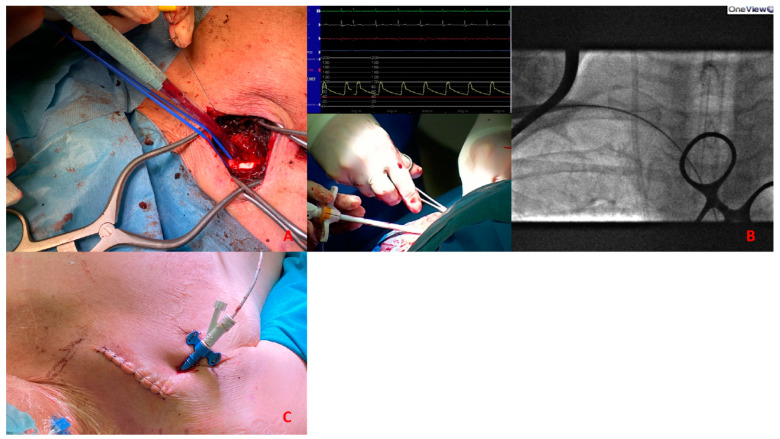
(**A**). Surgical exposure of subclavian artery. (**B**). Insertion of Impella CP sheath under fluoroscopy guidance. (**C**). Placement of the sheath after tunnellization under the skin.

**Figure 8 jpm-13-00293-f008:**
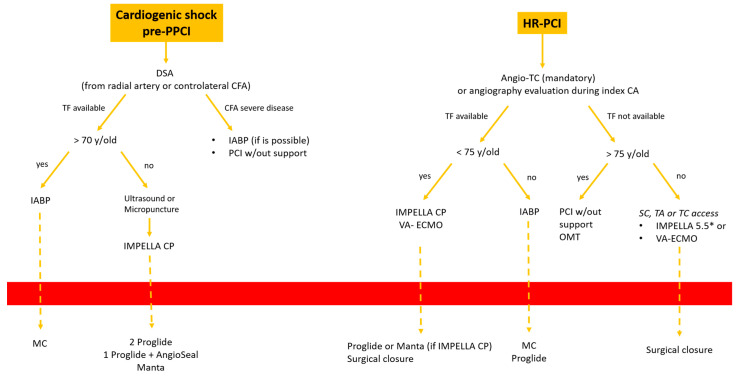
Decisional algorithm in setting of cardiogenic shock (CS) pre-PPCI and high risk-PCI. * Surgical cutdown.

**Figure 9 jpm-13-00293-f009:**
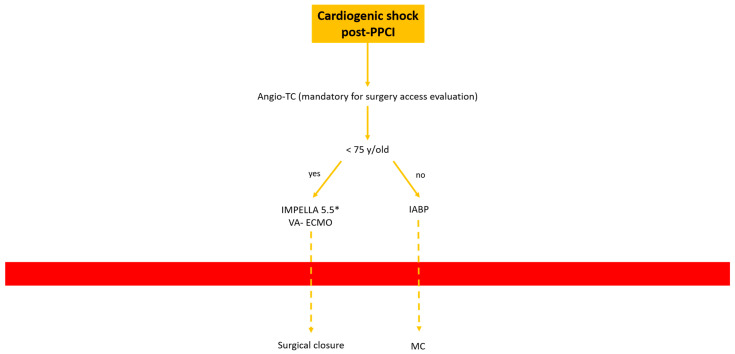
Decisional algorithm in setting of cardiogenic shock (CS) post-PPCI. * Surgical cutdown.

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
