# Peer review of "Management of Vascular Access in the Setting of Percutaneous Mechanical Circulatory Support (pMCS): Sheaths, Vascular Access and Closure Systems"

_jpm, 2023, doi:10.3390/jpm13020293_

Round 1

Reviewer 1 Report (Previous Reviewer 3)

Highlighting in red all changes in the revised version of the manuscript would have helped the revision process. However, the authors adequately addressed my concerns and manuscript improved significantly without the case report in it.

Author Response

Thanks to the reviewer for helping to improve our work

Reviewer 2 Report (New Reviewer)

This review summarizes pMCS selection, approaches, and hemostatic devices for a variety of situations, which we felt would provide useful information for real-world clinical practice.

There are several questions and corrections that need to be made, which are noted below.

The approach may differ depending on the circumstances of cardiogenic shock. In ECPR for out-of-hospital cardiac arrest, there is no time to perform angiography for puncture when inserting VA-ECMO, and echo-guided puncture would be the only option. Is it correct to say that the introduction of VA-ECMO in ECPR is outside of the three scenarios? Please describe this point.

I don't understand why the type of pMCS should be changed between patients over 70 and under 70, or over 75 and under 75. I can understand if the reason is that the access situation is different (e.g. shaggy aorta). (For example, IABP cannot be used because of shaggy aorta, etc.) Please describe the reason for changing the type of pMCS according to age.

If the patient can be weaned off a large diameter PMCS immediately after the procedure, pre-closure with proglide is useful. However, if the PMCS needs to be supported for a while after the procedure, either surgical hemostasis or manual compression hemostasis would be the choice. Please explain this point.

In the case of ECMO, perfusion of the lower extremity by progressive puncture of the sheath seems to be essential. Please include this information in your review.

Author Response

We thank the reviewer to highlight this important point:

1) It is a right comment.
Use of VA-ECMO in ECPR is outside of the three scenarios; that's why we describe the use of DSA as first option before to perform femoral puncture.

2) The reason for choosing pMCS with patient's age is related of the clinical practice of our centre and our protocols.

3) In our clinical practice we prefer to procede with pre-closure with proglide, mostly in the large bore access, in all the settings in which it is possible.
We always try to limit, when possible, the use of surgical hemostasis.

This manuscript is a resubmission of an earlier submission. The following is a list of the peer review reports and author responses from that submission.

Round 1

Reviewer 1 Report

The authors propose the article for the review section, but for this purpose it is necessary that the topics are treated in more detail and not as a simple list of devices or techniques.

Moreover, greater precision and accuracy is required in dealing with the various chapters.

The two specific and different clinical contexts highlighted in the article, namely high-risk coronary angioplasty (PCI) and cardiogenic shock / refractory cardiac arrest, involve substantial differences in the use of the different types of mechanical hemodynamic support systems and in the modalities of vascular approach, with differences in the outcome and incidence of complications. It is therefore essential to distinguish the two areas in the discussion of the topic and in the citation of the clinical trials of reference.

Currently, in high-risk PCI, mechanical hemodynamic support is aimed at preventing hemodynamic deterioration and the onset of cardiogenic shock during the procedure. Based on the risk stratification, IABP or Impella can be selected. For this indication there are scientific evidences that allow to orient in the choice of the patient. Prophylactic use of ECMO during PCI is generally considered a second choice.

In patients with cardiogenic shock, the failure of the systematic use of IABP has indeed opened the door to a wider use of Impella and ECMO.

In both clinical settings, the Tandem Heart system is little used in clinical practice.

The authors deal extensively with the most commonly used transfemoral vascular access, much less than the other vascular accesses; the anatomical approach (the use of palpable landmarks based on the iliac crest and pubic bone), although unreliable, is not mentioned; US-guided puncture of the common femoral artery is also poorly evaluated in favor of fluoroscopic guidance.

Finally, the examination of vascular hemostatic systems appears biased in favor of the Manta system and does not take into account the results of the systematic review of the Cochrane database on vascular closure devices for haemostasis of the femoral arterial puncture site and of the meta-analysis network. of studies comparing closure devices for femoral access after percutaneous coronary intervention.

The case report needs a more in-depth presentation: its classification in the current guidelines, the reasons for the deviation, if possible also a mention of the Revived clinical trial just presented at the ESC 2022 congress.

In particular:

-What vascular access was used for coronary angiography? In case of radial access, currently first choice in cath-labs, they should have already highlighted the obstruction of the subclavian artery.

- Better specify the type of revascularization performed and include post-PCI images of the right coronary artery; in fact from the pre-PCI images, it seems that there is a critical disease of the predivisional LM with the involvement of both the IVA ostium and the CX ostium; from the post-PCI images it appears that the LM-IVA-CX axis (T-stenting?) and the mid-segment of IVA and CX have been treated, no images of the RC artery are included.

Reviewer 2 Report

In the review, the authors provide a nice summary of available mechanical support device, basic vascular access techniques at different sites with valuable tips for safety and success. Later in the review, the authors provides a short summary of available vascular closure systems and a case report. 

Reviewer 3 Report

In the manuscript entitled "Management of vascular access in the setting of percutaneous mechanical circulatory support (pMCS): sheaths, vascular access and closure systems" Sardone et al described several aspects related to the management of vascular access in pMCS and described a case report from their centre experience.

This review is worthwhile reading to recap the current knowledge, since it summarizes significant aspects of this scenario.

> Nevertheless, some questions should be addressed:

1) Although the case report may be interesting to read, I do not see its value in a manuscript that is organized in this way, as a review. I would suggest using the case report as an example of what the authors want to highlight, rewording the title as "Management of vascular access in the setting of percutaneous mechanical circulatory support (pMCS): sheaths, vascular access and closure systems - "A case report and a review of literature". The manuscript would seem more attractive and the case report section may be more useful for the readers and I would strongly recommend it. In this case, I would rewrite the manuscript, giving the case report a "priority", moving the section after the introduction and I would rewrite it in a more attractive way. E.g. I would pose more "doubts" in the case report for the readers, first wondering why a closure system has been preferred over another one, anticipating the description of the different systems. Only in this way this section my be helpful for the readers, instead this section is overall useless.

2) Eventually, I would also add (if the authors have this possibility) different case reports (very briefly) using different closure systems, highlighting why the authors have preferred them over others, anticipating pros and cons of the different systems.

3) Also the description of the different systems should not be merely description, but should have a more clinical cut. An image summarizing the different systems in the different scenario, describing why a clinician should prefer a system over another is mandatory in a clinical review on the topic.

> Minor comments:

1) The manuscript should be reviewed by a native english speaker since in some paragraph is not fluent to read. Some sentences (and especially the abstract) would benefit from an entire rewording in order to convey messages in a more straightforward way.

2) Some typos should be corrected after a careful manuscript proofreading (e.g. "Intra-aortic balloon pump." - "Suture-based devices." > remove full stops after the titles ecc.)